# A randomised controlled trial of Pre-Operative Oncotype DX testing in early-stage breast cancer (PRE-DX study) – Study protocol

Matthew Northgraves[1]*, Judith Cohen[1], James Harvey[2,3], Chao Huang[1,4], Carlo Palmieri[5,6], Sarah Pinder[7,8], Pankaj Roy[9], Sarah Reynia[10], Marta Soares[11], Henry Cain[12]

1 Hull Health Trials Unit, University of Hull, Hull, United Kingdom, 2 Nightingale Breast Centre, Manchester University NHS Foundation Trust, Manchester, United Kingdom, 3 Division of Cancer Sciences, Faculty of Biology, Medicine and Health, University of Manchester, Manchester, United Kingdom, 4 Institute for Clinical and Applied Health Research (ICAHR), University of Hull, Hull, United Kingdom, 5 Institute of Systems, Molecular and Integrative Biology, Molecular and Clinical Cancer Medicine, University of Liverpool, Liverpool, United Kingdom, 6 The Clatterbridge Cancer Centre NHS Foundation Trust, Liverpool, United Kingdom, 7 School of Cancer & Pharmaceutical Sciences, Faculty of Life Sciences and Medicine, King's College London, London, United Kingdom, 8 Department of Cellular Pathology, Guy's and St Thomas NHS Foundation Trust, London, United Kingdom, 9 Department of Breast Surgery, Oxford University Hospitals NHS Foundation Trust, Oxfordshire, United Kingdom, 10 Exact Sciences UK, London, United Kingdom, 11 Centre for Health Economics, University of York, York, United Kingdom, 12 Royal Victoria Infirmary, Newcastle upon Tyne Hospitals NHS Foundation Trust, Newcastle upon Tyne, United Kingdom

* matthew.northgraves@hyms.ac.uk

**Data Availability Statement:** No datasets were generated or analysed during the current study. All

## Abstract

### Background

The Oncotype DX® Breast Recurrence Score assay can guide recommendations made to patients with oestrogen receptor positive (ER+), human epidermal growth factor receptor 2 negative (HER2-) breast cancer regarding post-surgery adjuvant therapy. Standard practice is to order the test in the post-operative setting on a specimen from the excised invasive carcinoma. However, it has been shown to be technically possible to perform the test on the diagnostic core biopsy. By testing the diagnostic core biopsy in the pre-operative setting, the wait for excised invasive carcinoma Recurrence Score results could be reduced allowing patients to be more accurately counselled regarding their treatment pathway sooner with any adjuvant treatment recommendations expedited. This would allow for more efficient streaming of follow up appointments. The aim of this study is to compare the impact on the patient treatment pathway of performing the Oncotype DX® test on the diagnostic core biopsy pre-operatively (intervention) as opposed to the excised invasive carcinoma (control).

### Methods and analysis

This parallel group randomised controlled trial aims to recruit 330 newly diagnosed patients with grade 2 or grade 3, ER+, HER2-, invasive intermediate risk early-stage breast cancer. Participants will be randomised 2:1 to the preoperative testing of the diagnostic core biopsy compared to the post-operative testing of the excision specimen. The primary endpoint is

relevant data from this study will be made available upon study completion.

**Funding:** The study is funded by a grant from Exact Sciences as an Investigator Initiated Trial. The funders had and will not have a role in study design, data collection and analysis, decision to publish, or preparation of the manuscript

**Competing interests:** Matthew Northgraves, Chao Huang and Marta Soares have nothing to declare. Judith Cohen reports a grant from commercial funder Exact Sciences for the conduct of the study. Henry Cain reports a grant for the conduct of this study as well as, honoraria for presentations/ educational events, and consulting fees from Exact Sciences. James Harvey reports a consultancy fee from Genomic Health. Carlo Palmieri reports consulting fees from Exact Sciences along with AstraZeneca, Daiichi Sankyo, Eli Lilly, Eisai, Gilead, Medac, MSD, Novartis, Pfizer, Roche, Seagen; grants from Daiichi Sankyo, Gilead, Pfizer, Seagen; honoraria for presentations/educational events from Pfizer, AZ, Seagen; receipt of equipment from Pfizer, Seagen; travel and meeting support from Gilead, Roche, Novartis and a role with the NCRI. Sarah Pinder reports honoraria for presentations/ educational events, travel support and participation on advisory boards and workshops for Exact Sciences. Pankaj .G. Roy reports personal fees from Exact Sciences, outside the submitted work. Sarah Reynia reports employment from, and stock in Exact Sciences. This does not alter our adherence to PLOS ONE policies on sharing data and materials.

number of clinical touchpoints between treating team and patient from initial approach until offer and prescription of the first adjuvant treatment. Secondary endpoints include time from diagnosis to offer and prescription of the first adjuvant treatment, patient-reported anxiety scores and health cost impact analysis collected at baseline, following the post-operative clinic and following the offer of adjuvant treatment, and number of alterations in treatment sequence from original planned surgical treatment to neoadjuvant therapy.

## Trial registration

The study was registered on ISRCTN (ISRCTN14337451) on the 16[th] August 2022.

## Introduction

Oncotype DX® is a 21-gene genomic assay used in early oestrogen receptor positive (ER+), human epidermal growth factor receptor 2-negative (HER2-) breast cancer [1]. The test has been validated as both a predictive and prognostic tool to estimate a patient's risk of distant breast-cancer recurrence. The use of the Recurrence Score (RS) result guides recommendations made to the patient regarding the requirement for adjuvant chemotherapy in addition to adjuvant endocrine therapy [2] with its use approved by the National Institute of Care and Excellence [3].

Traditionally the Oncotype test was performed on the surgical excision specimen following surgical treatment, however during the COVID-19 pandemic treatment recommendations changed as hospitals looked to defer surgery in preference for neoadjuvant endocrine therapy in appropriate patients [4]. To help inform treatment decisions, Oncotype testing on the core biopsy was used. It has been shown to be both technically possible and reliable to perform the Oncotype DX® test on the diagnostic core biopsy of the breast cancer prior to the surgical tumour excision procedure [5], potentially improving the timeliness of starting adjuvant treatment [6].

The availability of the Recurrence Score results in the pre-operative setting prior to the post-operative multidisciplinary team meeting has the potential to benefit the treating clinician, patient, and overall hospital cancer services. The treating clinician may be able to counsel the patient more accurately regarding their post-operative adjuvant treatment pathway at an earlier point in their pathway. From a patient perspective, understanding their complete cancer treatment pathway at an earlier stage may well reduce the anxiety associated with the treatment of their disease.

A need for a streamlining of multidisciplinary team (MDT) meetings in cancer care has been identified as demand for cancer services have increased [7]. The Independent Cancer Taskforce Report in 2015 recommended that NHS England encourage providers to focus specialist time in the MDT meetings on those cases which do not follow well-established clinical pathways. The pre-operative knowledge regarding the type of oncological input required in the adjuvant setting could potentially expedite those treatments by eliminating the inherent potential delay in waiting for the RS results on the surgical excision specimen. This would allow the streamlining of follow-up appointments and mitigate against unnecessary costly outpatient attendances.

### Aims and objectives

The aim of the study is to assess the impact on the patient management pathway by obtaining the Recurrence Score from the diagnostic core biopsy in the pre-operative setting, compared to the standard patient pathway, whereby the Oncotype DX test is requested post-operatively on the excision specimen.

### Study hypothesis

The primary hypothesis is that by having the RS results from the core biopsy rather than waiting for the RS on the excision specimen, the number of unnecessary outpatient appointments will be reduced, streamlining the patient management pathway.

The secondary hypotheses are that the availability of the RS results from the pre-operative core biopsy will reduce healthcare utilisation, improve patient experience, and decrease the time to the offering of relevant adjuvant cancer therapy.

## Methods and analysis

### Study design

PRE-DX is a multi-centre, prospective parallel group, randomised controlled trial with unequal random allocation ratio of 2:1 (intervention: standard care), using computer generated random permuted blocks, stratified by NHS centre and disease staging. A statistician, independent from the study team, will prepare the allocation schedule. The study is unblinded, with the research teams and participants aware of the study arm to which they have been allocated. The study is comparing the impact on the patient management pathway of performing the Oncotype DX test on the diagnostic core biopsy in the pre-operative setting (intervention) as opposed to the surgical excision specimen (control/standard). The primary outcome is the number of clinical touch points between diagnosis until the offer and prescription of adjuvant treatment. We will include a cost impact analysis to evaluate the health system costs from any potential streamlining of the patient management pathway.

The SPIRIT [8] schedule of events is displayed in Fig 1 and the completed SPIRIT checklist is in S1 Checklist.

### Study setting

We will look to recruit from up to 25 participating NHS centres across the UK.

### Eligibility criteria

Newly diagnosed patients with grade 2 or grade 3, ER+, HER2-, invasive intermediate risk early-stage breast cancer planned for surgery as their first definitive treatment will be recruited. Patients receiving endocrine bridging therapy are eligible for the study provided the reason for receipt is due to a delay in the planned surgery rather than for the purpose of downstaging the tumour. Potential participants will be identified during the diagnostic MDT meeting following a GP referral or a breast screening result. The diagnostic screening tests will be discussed during the MDT meeting with the diagnosis and disease staging confirmed and patient management plan determined. At this point the patient will be assessed against the full eligibility criteria for the study, which is presented in Table 1.

| TIMEPOINT** | STUDY PERIOD | | | | | | |
| --- | --- | --- | --- | --- | --- | --- | --- |
| | Enrolment | Allocation | Post-allocation | | | | |
| | *Pre-randomisation* | Randomisation | *Pre-operative Oncotype test ordered* | *Surgery* | *Post-op clinic appointments* | *Post-operative Oncotype test ordered* | *Offer of adjuvant treatment* |
| **ENROLMENT:** | | | | | | | |
| **Eligibility screen** | X | | | | | | |
| **Informed consent** | X | | | | | | |
| **Baseline questionnaire** | X | | | | | | |
| **Allocation** | | X | | | | | |
| **INTERVENTIONS:** | | | | | | | |
| *Oncotype DX performed preoperatively* | | | X | | | | |
| *Oncotype DX performed post operatively* | | | | | | X | |
| **ASSESSMENTS:** | | | | | | | |
| **Clinical touchpoints** | | | | X | X | | X |
| **Surgery histology eCRF** | | | | X | | | |
| **Hospital Anxiety and depression scale** | X | | | | X | | X |
| **Health resource Utilization questionnaire** | | | | | X | | X |

**Fig 1. PRE-DX Schedule of enrolment, interventions, and assessments.**

## Interventions

Standard practice is for the Oncotype DX® test to be ordered in the post-operative setting on a pathological specimen from the excised invasive carcinoma. The test was initially developed on cancer excision specimens, and this formed the basis of the NICE guidance [3]. This ordering of the test usually occurs around the time of the post-operative MDT meeting and post-

**Table 1. Study inclusion criteria.**

| Patients are eligible for inclusion if they are: |
| --- |
| • Male or female ≥ 18 years of age |
| • Oestrogen receptor positive, HER2 negative invasive early-stage breast cancer confirmed on diagnostic core biopsy, as per definition of hormone receptor status in NICE DG34 guidance |
| • Pre-operative staging of grade 2 cancer ≥ 20 mm on all imaging modalities or grade 3 cancer of any size and N0 on axillary staging or any size or grade which is N1 on preoperative axillary ultrasound scan +/- Fine Needle Aspiration or core biopsy |
| • Surgery planned as first definitive treatment |
| • Fit for adjuvant chemotherapy |
| • Able to provide written or remote (eConsent or postal) informed consent |
| • Able to complete questionnaires and study assessments |
| * If local funding is available for oncotype testing on N1 patients |
| Patients will be excluded if they are: |
| • ER negative or HER2 positive breast cancer |
| • N2 disease on pre-operative staging |
| • Planned for neo-adjuvant systemic therapy |
| • Unfit for surgical treatment or systemic chemotherapy |
| • Are unable to provide informed consent |
| • Have co-existing malignant disease only if this would affect the study in the investigator's opinion |
| • Are unable to complete study questionnaires even with the assistance of the study nurse |
| • Are already participating in another clinical trial |

operative results clinic appointment. This approach has the advantage of pathological stage being available prior to ordering the test, avoiding the possibility of the test being ordered on patients who turn out not to fit with guideline to use. Participants randomised to standard practice will act as the control arm, with sites following local procedure for ordering and acting on the RS result. If a participant has received endocrine bridging therapy prior to surgery, due to potential impact on the RS result, those in the standard arm will have the test performed on the diagnostic core biopsy, but it will be ordered in the post-operative setting.

This will be compared to the intervention arm, whereby the Oncotype DX test will be performed on the core biopsy taken for diagnosis and reported histopathologically as invasive carcinoma in the pre-operative setting. The intention is for the RS results and excision pathology to be available to support decision-making on adjuvant treatment options at the first post-operative MDT meeting.

It is not intended for the study to change the participant's primary treatment pathway but rather to inform the post-surgery adjuvant treatment decisions. The interpretation of the RS result and treatment recommendations will remain at the discretion of the MDT and treating clinicians following national guidelines. If the RS results are received prior to surgery, it will be left to the Centre's discretion whether to discuss the results with the patient before surgery, but the intentions of the centre will be recorded.

If the decision is made to change the treatment pathway, with the patient subsequently receiving neoadjuvant chemotherapy or neoadjuvant endocrine therapy prior to surgery, this will be recorded. For those patients, clinical touchpoints, Hospital Anxiety and Depression Scale (HADS) and Health Resource Use Questionnaire (HRUQ) will no longer be collected, only the final post-operative pathology report at the completion of treatment.

## Outcomes

**Primary outcome (Number of clinical touchpoints).** The primary outcome is the number of interactions (touchpoints) between the treating team and the participant, from diagnostic clinic appointment to the offer and prescription of the first adjuvant cancer treatment. Touchpoint zero is the diagnostic clinic and is not included. All other subsequent participant-clinician interactions related to breast cancer treatment will be recorded until adjuvant treatment is offered and prescribed. These include the date of surgery, post-operative results clinic and any planned / unplanned phone calls or outpatient clinic appointments relating to breast care treatment (Table 2). The pre-operative assessment to confirm fitness for surgery is not included.

**Secondary outcomes.** *Time to the offer and prescription of first adjuvant cancer treatment.* The difference in days between the date of attendance at first diagnostic clinic and the offer and prescription of adjuvant cancer treatment will be recorded between the two groups.

*Time to start of first adjuvant cancer treatment.* The difference in days between the date of attendance at first diagnostic clinic and the start of the first adjuvant cancer treatment will be recorded between the two groups.

*Alteration in recommended treatment sequence from original planned surgery to neoadjuvant therapy.* The number of occurrences in each group when the treatment sequence is changed, such that the participant receives neoadjuvant treatment rather than proceeding directly to surgery, potentially due to the RS results from the core biopsy will be recorded.

*Patient anxiety and depression.* Participant reported anxiety and depression will be recorded at three points (baseline, following the post-operative result clinic and following the offer of adjuvant treatment). This will be measured using the HADS, which was originally developed by Zigmond and Snaith [9] to measure anxiety and depression in a general medical population of patients but has now become a well validated measure within health research [10]. In the event the post-operative result clinic and offer of adjuvant treatment occur at the same appointment (e.g., patient prescribed endocrine therapy at the post-operative clinic appointment), the HADS will be completed only once, and the results used for both time points.

*Correlation of preoperative staging with postoperative pathological staging.* A discordance between the clinical and pathological stage in around a third of cases has previously been reported [11]. Therefore, the correlation between preoperative clinical diagnostic staging with the postoperative pathological staging for tumour size and nodal status (according to TNM disease staging) will be measured.

*Rate of failure of the Oncotype DX® assay (RS result cannot be issued) on diagnostic core biopsy specimen.* We will record the number of occasions when the RS is unable to be issued on the core biopsy. The failure rate for issuing the RS on the excision specimen is reported to be <3% compared to <6% on the core biopsy sample [5]. In the event of a failure to issue the

**Table 2. Example of clinical touchpoints.**

| Hospital reported Participant-Clinician Interactions–primary outcome (Includes both face to face and virtual, planned and unplanned) |
| --- |
| Oncology clinician outpatient visits or equivalent |
| Oncology nursing outpatient visits or equivalent |
| Surgical clinician outpatient visits or equivalent (Including date of surgery) |
| Surgical nursing outpatient or equivalent |
| Radiology clinician outpatient visits or equivalent |
| Radiology nursing outpatient visits or equivalent |
| Other breast care related secondary care visits |

RS result on core biopsy, the excision specimen will be retested, as per local guidance but the participant will remain in the group originally allocated.

## Recruitment

Potential participants identified at the diagnostic MDT meeting will have the study discussed with them during the diagnostic clinic appointment where they are informed of their cancer diagnosis. If interested, the patient will be provided with the Participant Information Sheet (PIS) and referred to the site research team to discuss the study further. When any results required to confirm eligibility are outstanding, such as the HER2 status, the study can still be introduced during the diagnostic clinic appointment on the understanding the patient is aware their eligibility is dependent on the outstanding results. If the patient is not approached during the diagnostic clinic appointment, the study information (e.g. PIS) and invitation letter can be posted to the patient to be discussed at a future appointment.

The study documents are not being translated to other languages; however, the use of hospital translation services is permitted to support inclusion from all demographic groups. Reasons for ineligibility, or participants declining, are being recorded by sites on the study screening logs and closely monitored by Hull Health Trials Unit (HHTU).

## Consent

Once the patient's eligibility has been confirmed, the delegated site research team member will telephone the potential participant to answer any questions the patient might have. The call will take place no sooner than 24 hours after the patient has been provided with the PIS. There is no upper limit on how long a participant can consider participating in the study, beyond the fact that informed consent must be taken before surgery. If patient remains interested in participating, once the study has been fully explained, the authorised site team member can proceed to obtain informed consent. Consent will be completed by the study clinician, research nurse or any other appropriately trained individuals authorised to do so on the delegation log.

The consent process will primarily be completed using the remote consent methods of e-consent via DocuSign® or postal consent, as it is not intended for participants to attend the hospital outside of routine appointments. Consent will be taken following a real-time phone or videocall discussion with patient's identity checked and verified at the next face to face hospital appointment. For the study, if postal consent is used, the date that the patient has confirmed they have signed the consent form will be taken as the date of consent. In-person consent is appropriate if the patient is attending the hospital for a scheduled clinical appointment. The remote consent process is depicted in Fig 2. The consent includes the use of routinely stored archival tissue for future approved studies.

Copies of all completed consent forms will be uploaded into the study database for monitoring purposes and the participant's GP and clinical care team will be notified of their participation.

Once consented, the participant will complete the baseline assessment, which includes the HADS questionnaire, before being randomised by the authorised individual of the site team.

## Participant timeline

There will be no change to standard preoperative investigations or clinical appointments as scheduled by the treating clinical team according to site clinical care pathway protocols. There are no additional visits for the purpose of the study.

Data on the number of clinical touchpoints/visits and the total number of days from the patient diagnosis to the offer and prescription of the first adjuvant cancer treatment in both

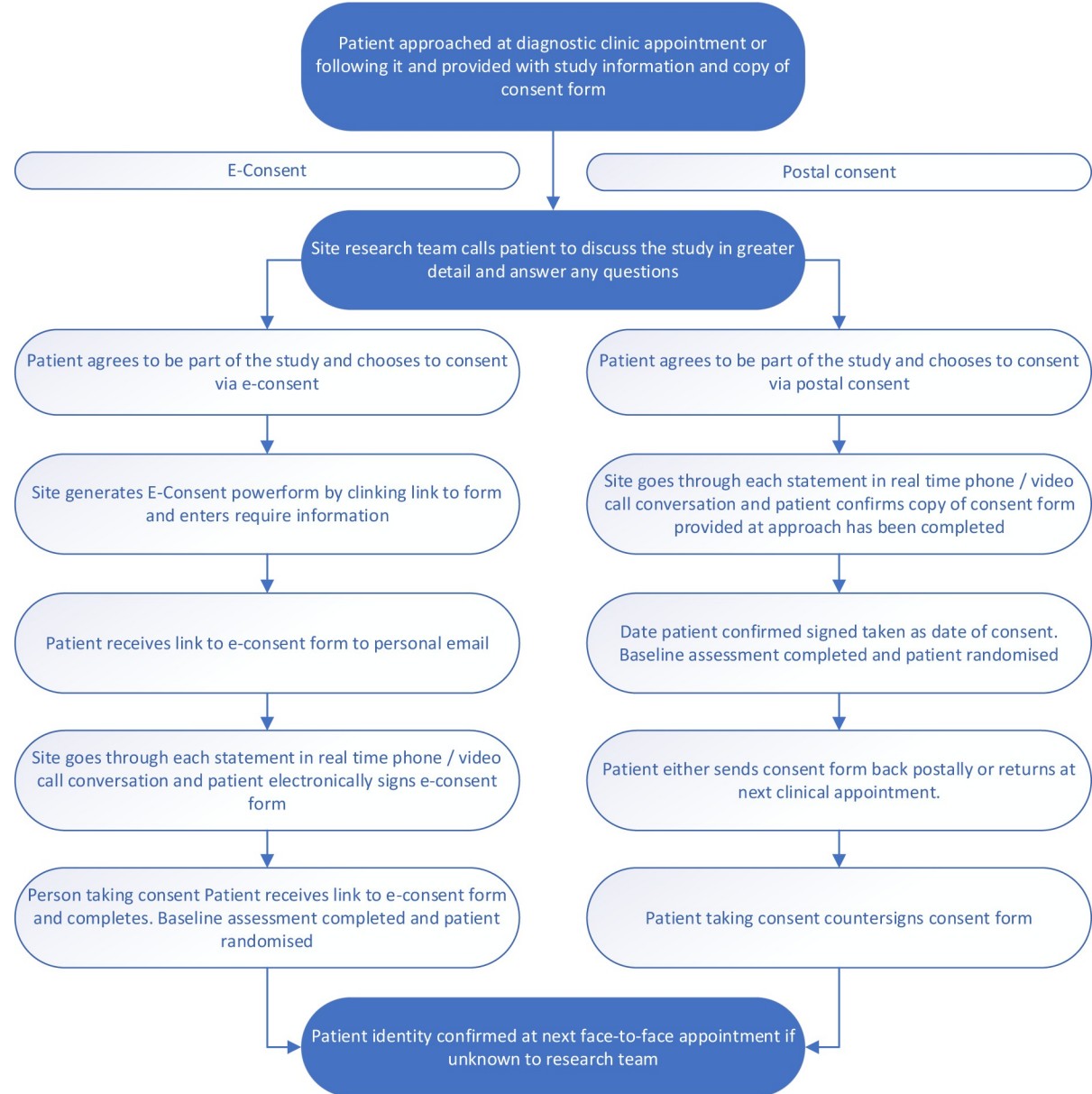

**Fig 2. Flow diagram of remote consent process.**

groups will be collected as a study outcome. At baseline, participants will complete the HADS questionnaire as part of the baseline assessment. The HADS and a HRUQ will then be completed following the post-operative clinic and following the offer of adjuvant treatment. In the event of the two follow-up events being the same date, only the post-operative clinic questionnaires will be completed.

## Sample size

The primary outcome is an analysis of the number of participant-clinician touch points. We hypothesise that the number of participant-clinician touch points in the control group is 6 visits per patient and our intervention is to reduce it to 5 visits per patient. With a 2:1 recruitment

ratio and 20% dropout rate, to detect a 1 touchpoint reduction between intervention and standard arm using a negative binomial model, at 5% significance level and 90% power, a sample size of 330 in total is required (220 in the intervention arm and 110 in the standard arm).

## Data collection methods

The majority of data collection will be directly into the electronic Case Report Forms (eCRFs) in the study database. Routinely collected hospital data will be recorded directly from the medical records with the option to complete paper CRFs which will then be transcribed into the eCRFs.

Follow-up patient reported outcomes (HADS and HRUQ) will be collected using three methods, depending on patient preference. Participants will be provided with the option of either receiving the follow-up questionnaires directly from the database by email, completing them over the phone with the delegated site team or by postal questionnaire sent by the site.

Sites will aim to send the questionnaires within 3 days; and no later than 7 days from the first post-operative clinic appointment date. If no response has been received within one week ± 3 days of the questionnaire being sent, the patient will be contacted by the site team by phone.

## Data management

The study database will be hosted and managed within the HHTU instance of commercial online data capture and randomisation system, REDCap Cloud (RCC). The database will be built and validated according to study specific requirements with automated checks and manual checks to monitor study data quality and completeness, as detailed in the Data Monitoring Plan. Missing data will be chased until it is received or confirmed as unavailable.

The e-consent information will be collected using Docusign® Powerforms, located within a sub-account of the HHTU instance of e-signature platform DocuSign®.

All the information obtained about participants during the study will be kept strictly confidential and will be held in accordance with the General Data Protection Regulation (GDPR 2018). In principle, the anonymised data generated by the study will be suitable for sharing, e.g. for further research analysis and meta-analysis. Final anonymised clinical study datasets and meta-data will be produced by HHTU data team and stored in an appropriate format to enable discoverability and sharing on The University of Hull data repository, Worktribe. Requests for access to the dataset will be managed via HHTU and the CI. If access to data is approved, a Data Sharing Agreement will be signed.

## Statistical analysis

The trial will be reported in accordance with the CONSORT 2010 statement [12] with a detailed statistical analysis plan provided by the trial statistician and approved by the Trial Management Group (TMG) prior to the data collection being completed. All statistical analyses will be performed using SPSS version 28 and R language version 4.2. Ninety-five percent confidence intervals will be presented with the significance of p value at 5% significance level. Analyses will be conducted on an intention-to-treat basis. Baseline patient characteristics will be tabulated through descriptive statistics.

The analysis of primary outcome, i.e. comparison of the numbers of participant-clinician touch points, will be undertaken using a negative binomial model, adjusting for stratification factors (hospital site and disease staging) and prognostic baseline covariates (grade and nodal status). The frequency of alteration in treatment pathway, e.g. whereby the participant was recommended neoadjuvant treatment, and the rate of failure of obtaining a RS result on

diagnostic core biopsy specimen in the intervention arm will be summarised. A per-protocol analysis will be conducted as secondary analysis, excluding those with alteration in treatment pathway or failure to obtain a RS result.

For the secondary outcomes, comparison of time between diagnosis and adjuvant treatment will be assessed via Cox proportional hazards regression, controlling for the same stratification factors and prognostic covariates as the primary outcome analysis. The analysis of HADS scores (at the post-operative result clinic and offer and prescription of adjuvant treatment) will be undertaken by linear regression, adjusting for the baseline HADS score, together with the same stratification factors and prognostic covariates. Correlation of preoperative staging with postoperative pathological staging will be calculated using Spearman's rank correlation.

There are no planned interim analyses for the trial or stopping guidelines for the study.

## Health system cost impact analysis

The health system impact analysis will evaluate the health system costs between the groups, including any potential streamlining of the patient management pathway due to the availability of the RS results from the core biopsy in the pre-operative setting. Patient reported health resource utilisation related to their breast care will be collected following the post-operative clinic appointment and following the offer of adjuvant treatment using a self-administered HRUQ. This will include both primary and secondary care and patient diaries will be provided to aid with appointment recall. Data on the number of health care provider-participant touch points throughout the follow-up, and the number of repeated tests will be collected. This will include cases where there is failure of the Oncotype DX assay to issue a result on the core biopsy specimen and re-testing proceeds using the surgical specimen.

The patient reported HRUQ will aim to identify any further impacts on health care resource use, e.g. relating to additional contacts with primary care providers. The resource utilisation will then be costed using appropriate sources, including NHS reference costs and national tariffs. The difference in resource use between the two arms of the trial will be descriptively summarised with any variation in total costs examined by linear regression, controlling for the same stratification factors and prognostic covariates as the statistical analyses.

## Auditing

The sponsor for the study is Newcastle upon Tyne Hospitals NHS Foundation Trust with the day-to-day management conducted by the HHTU. The study will be conducted in accordance with the principles of Good Clinical Practice as applicable under UK regulations, the NHS Research Governance Framework, and through adherence to Standard Operating Procedures (SOPs). The HHTU study team will oversee study monitoring activities and ensure that the study is conducted in line with agreed SOPs, including conducting internal audits on study management at least annually.

The TMG will review trial processes and procedures, including monitoring the rate of participant recruitment, data collection and any safety issues. As the study is looking at the effect of rescheduling the Oncotype DX test rather than there being any study specific assessments or altering the patient treatment, there will be no formal safety data collected. This is based on the study risk assessment. The TMG will monitor diagnosis and treatment outcomes and if any concerns about patient safety are identified then the TMG will be responsible for investigating and establishing any requirement for a change in study design and safety reporting procedures.

If the risk profile of the study changes during the study the need for an Independent Trial Steering Committee or Data Monitoring Committee will be reviewed.

## Regulatory approvals

NHS REC (Reference: 22/LO/0421; London - Surrey Research Ethics Committee) and HRA approval were granted on 26th July 2022. Any amendments will be submitted to the REC and HRA having been agreed with the sponsor and funder and classified according to HRA guidance. Amendments will be implemented at participating NHS organisations in agreement with the guidance and approval of the HRA and subject to confirmation of local capacity and capability.

## Current trial status

The study opened to recruitment on the 10[th] October 2022 with the first participant recruited on the 25[th] October 2022. The recruitment period is expected to last 12 months.

The current protocol version is V2.2 (27.06.2023) (S1 Protocol). One substantial amendment has been submitted since the original version was approved by the REC. This contained clarifications to the primary outcome and the consent process. In addition, the minimum size requirement for histological grade 3 tumours of $\geq$10mm has been removed, to replicate clinical practice. A clarification regarding the inclusion of patients receiving endocrine bridging therapy was included in non-substantial amendment 4.

## Dissemination

The study results will be disseminated through presentations, conferences, and peer-reviewed journals according to the PRE-DX publication and dissemination policy. The International Committee of Medical Journal Editors criteria [13] will be adhered to when determining authorship and contributions.

## Discussion

The study will add to the evidence base as to whether performing the Oncotype DX test on the diagnostic core biopsy in the preoperative setting can inform and improve the management pathway of early breast cancer for both the patient and the wider health care system.

## Supporting information

**S1 Checklist. SPIRIT checklist.**
(DOC)

**S1 Protocol. Approved protocol.**
(PDF)

## Author Contributions

**Conceptualization:** Matthew Northgraves, Judith Cohen, James Harvey, Chao Huang, Carlo Palmieri, Sarah Pinder, Pankaj Roy, Sarah Reynia, Marta Soares, Henry Cain.

**Formal analysis:** Chao Huang, Marta Soares.

**Funding acquisition:** Henry Cain.

**Investigation:** Matthew Northgraves.

**Methodology:** Matthew Northgraves, Judith Cohen, James Harvey, Chao Huang, Carlo Palmieri, Sarah Pinder, Pankaj Roy, Henry Cain.

**Project administration:** Matthew Northgraves.

**Supervision:** Henry Cain.

**Writing – original draft:** Matthew Northgraves.

**Writing – review & editing:** Matthew Northgraves, Judith Cohen, James Harvey, Chao Huang, Carlo Palmieri, Sarah Pinder, Pankaj Roy, Sarah Reynia, Marta Soares, Henry Cain.

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
