## [Decision Letter · Decision Letter 0]

4 Jan 2024

PONE-D-23-30937A Randomised Control trial of Pre-Operative Oncotype DX® Testing in early-stage breast cancer  – Study ProtocolPLOS ONE

Dear Dr. Northgraves,

Thank you for submitting your manuscript to PLOS ONE. After careful consideration, we feel that it has merit but does not fully meet PLOS ONE’s publication criteria as it currently stands. Therefore, we invite you to submit a revised version of the manuscript that addresses the points raised during the review process.

We look forward to receiving your revised manuscript.

Kind regards,

Daniele Ugo Tari, M.D.

Academic Editor

PLOS ONE

Journal Requirements:

"Matthew Northgraves has nothing to disclose.

 Judith Cohen reports a grant from Exact Sciences, during the conduct of the study.

 James Harvey reports a consultancy fee from Genomic Health.

 Chao Huang has nothing to disclose.

 Carlo Palmieri reports grants from Daiichi Sankyo, Gilead, Pfizer, Seagen, consulting fees from  AstraZeneca, Daiichi Sankyo, Eli Lilly, Eisai, Exact Sciences, Gilead, Medac, MSD, Novartis, Pfizer, Roche,  Seagen, honoraria for   presentations/educational events from Pfizer, AZ, Seagen, receipt of equipment from  Pfizer, Seagen, travel and meeting support from Gilead, Roche, Novartis and a role with the NCRI. 

 Sarah Pinder reports honoraria for presentations/educational events, travel support and participation on  advisory boards and workshops for Exact Sciences.

 Pankaj .G. Roy reports personal fees from Exact Sciences, outside the submitted work.

 Sarah Reynia reports employment from, and stock in Exact Sciences.

 Marta Soares has nothing to disclose.

 Henry Cain reports a grant, honoraria for presentations/educational events, and consulting fees from Exact  Sciences."

We note that you received funding from a commercial sources: Daiichi Sankyo, Gilead, Pfizer, Seagen, AstraZeneca, Eli Lilly, Eisai, Exact Sciences, Medac, MSD, Novartis, Roche & Genomic Health.

Within this Competing Interests Statement, please confirm that this does not alter your adherence to all PLOS ONE policies on sharing data and materials by including the following statement: ""This does not alter our adherence to PLOS ONE policies on sharing data and materials.” (as detailed online in our guide for authors http://journals.plos.org/plosone/s/competing-interests).  

If there are restrictions on sharing of data and/or materials, please state these. Please note that we cannot proceed with consideration of your article until this information has been declared. 

6. We note that the original protocol that you have uploaded as a Supporting Information file contains an institutional logo. As this logo is likely copyrighted, we ask that you please remove it from this file and upload an updated version upon resubmission. 

Reviewers' comments:

Reviewer's Responses to Questions

**Comments to the Author**

1. Does the manuscript provide a valid rationale for the proposed study, with clearly identified and justified research questions?

Reviewer #1: Yes

Reviewer #2: Yes

Reviewer #3: Partly

2. Is the protocol technically sound and planned in a manner that will lead to a meaningful outcome and allow testing the stated hypotheses?

Reviewer #1: Yes

Reviewer #2: Partly

Reviewer #3: Yes

3. Is the methodology feasible and described in sufficient detail to allow the work to be replicable?

Reviewer #1: Yes

Reviewer #2: Yes

Reviewer #3: Yes

4. Have the authors described where all data underlying the findings will be made available when the study is complete?

Reviewer #1: No

Reviewer #2: Yes

Reviewer #3: Yes

5. Is the manuscript presented in an intelligible fashion and written in standard English?

Reviewer #1: Yes

Reviewer #2: Yes

Reviewer #3: Yes

6. Review Comments to the Author

You may also provide optional suggestions and comments to authors that they might find helpful in planning their study.

Reviewer #1: In this study protocol, a two-arm randomized (2:1) controlled multi-center trial is being proposed which aims to compare the impact on the patient treatment pathway of performing Oncotype DX test on the diagnostic core biopsy pre-operatively (intervention) to the excised invasive carcinoma (control). The primary endpoint is the number of clinical interactions between treating team and patient from initial approach until offer and prescription of the first adjuvant treatment.

Minor revisions:

1- Line 353: Clarify if Cox regression implies Cox proportional hazards regression.

2- Line 357: State the statistical methods that will be used for correlating preoperative staging with postoperative pathological staging.

3- Identify the software that will be used for the statistical analysis.

Reviewer #2: Article review (A Randomised Control trial of Pre-Operative Oncotype DX® Testing in early-stage breast cancer – Study Protocol, PONE-D--23-30937):

Summary

This is a protocol for a randomized control trial comparing the preoperative Oncotype testing of the core biopsy to the postoperative Oncotype testing of the surgical pathology specimen. Primary endpoint is number of clinical touchpoints. Secondary endpoints include time from diagnosis to adjuvant therapy, patient-reported anxiety scores and health cost impact analysis, and number of alterations in treatment sequence from original planned surgical treatment to neoadjuvant therapy.

Comments/Revisions

1. The impact of preopereative Oncotype testing in the management of early-stage breast cancer is worthy investigation and there is merit in conducting the proposed trial

2. The main benefit of preoperative testing would be the reduction in time from diagnosis to adjuvant therapy. Therefore, I feel that time to adjuvant therapy should be the primary endpoint. Number of clinical touchpoints can be one of the secondary endpoints.

3. The agreement between preoperative and postoperative Onctotype should be reported.

4. Professional editing is recommended prior to publication

Recommendation

Major revision

Reviewer #3: I am very grateful to you for giving me the opportunity to review this interesting manuscript where the authors describe a new approach to the treatment of early breast cancer. The authors propose to perform the OncotypeDx on the biopsy sample of patients with surgical indication, instead of surgical specimen, in order to optimize the initiation of adjuvant treatment. However, I have a few concerns about the protocol:

- The decision for adjuvant treatment is typically based on an evaluation of both clinical and genomic risk factors, as detailed by Sparano et al (N Engl J Med 2019; 380:2395-2405). Therefore, it is an interesting strategy to conduct the genomic analysis on the biopsy sample, ensuring that the initiation of adjuvant treatment is not delayed. Nevertheless, there are certain scenarios in which it is unnecessary to utilize the genomic platform to make decisions about adjuvant treatment. This includes cases involving premenopausal patients with positive axillary fine-needle aspiration (FNA) or tumors larger than 3 cm with grade 3. Performing genomic platforms in this context could potentially lead to unnecessary costs for the health system. I kindly suggest to the authors to consider and describe strategies in the protocol that could help minimize such scenarios during the study, with the aim of optimizing resource utilization.

- The authors describe the analysis of the number of contact points between the clinician and the participant as the primary outcome and calculate the sample size based on the reduction from 6 to 5 interactions. I kindly suggest that the authors provide an explanation in the manuscript regarding why this reduction might be considered relevant in the treatment of these patients. Additionally, the authors could discuss how this reduction might impact costs to the healthcare system. Providing such context would enhance the understanding of the significance of the chosen primary outcome and its implications for both patient care and resource utilization.

7. PLOS authors have the option to publish the peer review history of their article (what does this mean?). If published, this will include your full peer review and any attached files.

Reviewer #1: No

Reviewer #2: **Yes: **Ioannis Alagkiozidis

Reviewer #3: No

---

## [Author Response · Author response to Decision Letter 0]

6 Feb 2024

Response

We have corrected the author addresses as per the title page formatting guidance, removing the post code from affiliation address 9 (Pankaj Roy) – Lines 21-22, the street and building from affiliation address 10 (Sarah Reynia) – Line 23, and street from affiliation address 12 (Henry Cain) -lines 25-26.

The citing and presentation of titles for all figures have been corrected to comply with the formatting guidance for the main body of text.

In addition, the file naming has been corrected (Lines 119-120, 246, 249).

Response

We have removed the funding statement from the title page as per the title page formatting guidance. 

"Matthew Northgraves has nothing to disclose.

Judith Cohen reports a grant from Exact Sciences, during the conduct of the study.

James Harvey reports a consultancy fee from Genomic Health.

Chao Huang has nothing to disclose.

Carlo Palmieri reports grants from Daiichi Sankyo, Gilead, Pfizer, Seagen, consulting fees from AstraZeneca, Daiichi Sankyo, Eli Lilly, Eisai, Exact Sciences, Gilead, Medac, MSD, Novartis, Pfizer, Roche, Seagen, honoraria for presentations/educational events from Pfizer, AZ, Seagen, receipt of equipment from Pfizer, Seagen, travel and meeting support from Gilead, Roche, Novartis and a role with the NCRI. 

Sarah Pinder reports honoraria for presentations/educational events, travel support and participation on advisory boards and workshops for Exact Sciences.

Pankaj .G. Roy reports personal fees from Exact Sciences, outside the submitted work.

Sarah Reynia reports employment from, and stock in Exact Sciences.

Marta Soares has nothing to disclose.

Henry Cain reports a grant for the conduct of this study, honoraria for presentations/educational events, and consulting fees from Exact Sciences."

We note that you received funding from a commercial sources: Daiichi Sankyo, Gilead, Pfizer, Seagen, AstraZeneca, Eli Lilly, Eisai, Exact Sciences, Medac, MSD, Novartis, Roche & Genomic Health.

Within this Competing Interests Statement, please confirm that this does not alter your adherence to all PLOS ONE policies on sharing data and materials by including the following statement: ""This does not alter our adherence to PLOS ONE policies on sharing data and materials.” (as detailed online in our guide for authors http://journals.plos.org/plosone/s/competing-interests). 

If there are restrictions on sharing of data and/or materials, please state these. Please note that we cannot proceed with consideration of your article until this information has been declared. 

Response

The following amended Competing Interest Statement has been added to the cover letter. 

Matthew Northgraves, Chao Huang and Marta Soares have nothing to declare.

Judith Cohen reports a grant from commercial funder Exact Sciences for the conduct of the study.

Henry Cain reports a grant for the conduct of this study as well as, honoraria for presentations/educational events, and consulting fees from Exact Sciences."

James Harvey reports a consultancy fee from Genomic Health.

Carlo Palmieri reports consulting fees from Exact Sciences along with AstraZeneca, Daiichi Sankyo, Eli Lilly, Eisai, Exact Sciences, Gilead, Medac, MSD, Novartis, Pfizer, Roche, Seagen; grants from Daiichi Sankyo, Gilead, Pfizer, Seagen; honoraria for presentations/educational events from Pfizer, AZ, Seagen; receipt of equipment from Pfizer, Seagen; travel and meeting support from Gilead, Roche, Novartis and a role with the NCRI.

Sarah Pinder reports honoraria for presentations/educational events, travel support and participation on advisory boards and workshops for Exact Sciences.

Pankaj .G. Roy reports personal fees from Exact Sciences, outside the submitted work.

Sarah Reynia reports employment from, and stock in Exact Sciences.

This does not alter our adherence to PLOS ONE policies on sharing data and materials.

Response

The manuscript title has been amended to ‘A Randomised Controlled trial of Pre-Operative Oncotype DX® Testing in early-stage breast cancer – Study Protocol’ (Line 1) and the online submission title has been edited accordingly so it now matches the manuscript as requested. 

Response

Supporting information captions have been added to the end of the manuscript (Lines 426-428) and in-text citations included (lines 119-120 and 373) as requested. 

6. We note that the original protocol that you have uploaded as a Supporting Information file contains an institutional logo. As this logo is likely copyrighted, we ask that you please remove it from this file and upload an updated version upon resubmission. 

Logos in the Original REC approved protocol have been removed as requested. 

Reviewer One comments:

Reviewer #1: In this study protocol, a two-arm randomized (2:1) controlled multi-center trial is being proposed which aims to compare the impact on the patient treatment pathway of performing Oncotype DX test on the diagnostic core biopsy pre-operatively (intervention) to the excised invasive carcinoma (control). The primary endpoint is the number of clinical interactions between treating team and patient from initial approach until offer and prescription of the first adjuvant treatment.

Minor revisions:

1- Line 353: Clarify if Cox regression implies Cox proportional hazards regression.

Response

We can confirm that Cox proportional hazards regression will be used and the manuscript has been updated accordingly (line 320). 

2- Line 357: State the statistical methods that will be used for correlating preoperative staging with postoperative pathological staging.

Response

We will use Spearman’s rank correlation for correlating preoperative staging with postoperative pathological staging. The manuscript has been updated accordingly (lines 325-326). 

3- Identify the software that will be used for the statistical analysis.

Response

All statistical analyses will be performed using SPSS version 28 and R language version 4.2. The manuscript has been updated accordingly (lines 306-307). 

Reviewer two comments:

Comments/Revisions

1. The impact of preoperative Oncotype testing in the management of early-stage breast cancer is worthy investigation and there is merit in conducting the proposed trial

Response

We would like to thank the reviewer for agreeing the topic is worthy of investigation and that there is merit in performing the proposed trial. 

2. The main benefit of preoperative testing would be the reduction in time from diagnosis to adjuvant therapy. Therefore, I feel that time to adjuvant therapy should be the primary endpoint. Number of clinical touchpoints can be one of the secondary endpoints.

Response

We agree that reducing the time from diagnosis to adjuvant treatment is one of the main benefits of pre-operative oncotype testing. This was given due consideration during the development stage of the study; however, it was felt the reduction of clinical touchpoint was a more suitable primary outcome measure for the following reasons:

a) The COVID-19 pandemic caused considerable backlogs in NHS appointments and waiting times grew. As a result, some NHS trusts implemented a standard two to three months wait time for post-surgery oncology appointments. Had this been a single site study we would had been more confident of using time to surgery but due to the variability across multiple units in terms of waiting list, it was felt this may have diluted any potential benefits that may have been seen. 

b) As discussed in the introduction, there is a large financial benefit to reducing the number of costly outpatients’ appointments. Current NHS National Tariff costs (2022/23 Annex A: The national tariff workbook) have consultant led follow-up Breast surgery outpatient appointment costing £79-£97 and clinical oncology outpatient at £132 - £147 per appointment. There would be a significant financial benefit to the NHS from removing unnecessary appointments whilst potentially reducing waiting times. 

c) Reducing the number of hospital visits from a patient’s point of view is very important and highlights its value as a primary endpoint. 

It would not be possible to change the primary endpoint at this point in the study, as recruitment and data collection are in progress. The trial was designed with the primary outcome as defined and the sample size calculated based on this. It is an important principle in the good conduct of clinical trials that the primary outcome is predefined before the study commences.

3. The agreement between preoperative and postoperative Oncotype should be reported.

Response

We wholeheartedly agree that the agreement between the preoperative and postoperative Oncotype is an important measure, however at the current time, we have not got the funding to enable analysis using paired sample testing. The Oncotype test was funded according to available NHS funding for clinical care and performed once at the time point dependent on randomised allocation. It would be possible to order the additional test on the pre or post-operative samples that are taken routinely, but the additional testing would need to be funded from the research funder. This element of the study has been presented to the funder as an additional sub-study and is still under consideration. A consent statement has also been included in the participant consent forms allowing the paired sample to be conducted if the funding is secured. 

4. Professional editing is recommended prior to publication

Response

We unfortunately do not have the budget for professional editing. 

Reviewer three comments:

Reviewer #3: I am very grateful to you for giving me the opportunity to review this interesting manuscript where the authors describe a new approach to the treatment of early breast cancer. The authors propose to perform the OncotypeDx on the biopsy sample of patients with surgical indication, instead of surgical specimen, in order to optimize the initiation of adjuvant treatment. However, I have a few concerns about the protocol:

1. The decision for adjuvant treatment is typically based on an evaluation of both clinical and genomic risk factors, as detailed by Sparano et al (N Engl J Med 2019; 380:2395-2405). Therefore, it is an interesting strategy to conduct the genomic analysis on the biopsy sample, ensuring that the initiation of adjuvant treatment is not delayed. Nevertheless, there are certain scenarios in which it is unnecessary to utilize the genomic platform to make decisions about adjuvant treatment. This includes cases involving premenopausal patients with positive axillary fine-needle aspiration (FNA) or tumors larger than 3 cm with grade 3. Performing genomic platforms in this context could potentially lead to unnecessary costs for the health system. I kindly suggest to the authors to consider and describe strategies in the protocol that could help minimize such scenarios during the study, with the aim of optimizing resource utilization.

Response

We agree with the reviewer’s comment that there are certain scenarios when clinically it is unnecessary to utilise the genomic platform as described. Recruiting sites were informed during the site set-up that only patients for whom the oncotype was clinically required to guide their treatment were eligible for the study. This message has been reinforced during the recruitment period at monitoring calls with sites. The test is ordered as part of the patients’ standard of care, and sites were informed that no tests should be ordered purely for the purpose of the research study. We acknowledge that there is the potential that tests ordered in the pre-operative setting may no longer be required in the post-operative setting (e.g. cancer discovered to be metastatic during surgery) and the frequency at which this occurs will be reported. 

As this detail was not explicitly stated in the REC approved protocol we do not think that it would be appropriate to include the details in this manuscript. However, this will be reported and discussed in the results paper. 

2. The authors describe the analysis of the number of contact points between the clinician and the participant as the primary outcome and calculate the sample size based on the reduction from 6 to 5 interactions. I kindly suggest that the authors provide an explanation in the manuscript regarding why this reduction might be considered relevant in the treatment of these patients. Additionally, the authors could discuss how this reduction might impact costs to the healthcare system. Providing such context would enhance the understanding of the significance of the chosen primary outcome and its implications for both patient care and resource utilization.

Response

Again, we thank the reviewer for these very valid suggestions. The reduction from 6 to 5 clinician and participant interactions came from pre-study modelling, based on typical pathways at the four breast units which have members on the trial management group. We agree discussion regarding why the reduction may be relevant to patients and more widely what the impact on resource utilization could be is important and will be included in the results paper. However, as stated above, as this detail was not explicitly stated in the REC approved protocol, we do not think that it would be appropriate to include the details in this manuscript.

---

## [Decision Letter · Decision Letter 1]

27 Feb 2024

A Randomised Controlled trial of Pre-Operative Oncotype DX® Testing in early-stage breast cancer (PRE-DX study) – Study Protocol

PONE-D-23-30937R1

Dear Dr. Northgraves,

We’re pleased to inform you that your manuscript has been judged scientifically suitable for publication and will be formally accepted for publication once it meets all outstanding technical requirements.

Kind regards,

Daniele Ugo Tari, M.D.

Academic Editor

PLOS ONE

Additional Editor Comments (optional):

Reviewers' comments:

Reviewer's Responses to Questions

**Comments to the Author**

1. Does the manuscript provide a valid rationale for the proposed study, with clearly identified and justified research questions?

Reviewer #1: Yes

Reviewer #2: Yes

Reviewer #3: Partly

2. Is the protocol technically sound and planned in a manner that will lead to a meaningful outcome and allow testing the stated hypotheses?

Reviewer #1: Yes

Reviewer #2: Yes

Reviewer #3: Yes

3. Is the methodology feasible and described in sufficient detail to allow the work to be replicable?

Reviewer #1: Yes

Reviewer #2: Yes

Reviewer #3: Yes

4. Have the authors described where all data underlying the findings will be made available when the study is complete?

Reviewer #1: No

Reviewer #2: Yes

Reviewer #3: Yes

5. Is the manuscript presented in an intelligible fashion and written in standard English?

Reviewer #1: Yes

Reviewer #2: Yes

Reviewer #3: Yes

6. Review Comments to the Author

You may also provide optional suggestions and comments to authors that they might find helpful in planning their study.

Reviewer #1: All comments have been adequately addressed.

Reviewer #2: Accepted. The authors have addressed the points made at the initial review and their proposed protocol meets standards for publication

Reviewer #3: No modifications have been made to the manuscript according to the comments sent. I consider that the two major comments made are related to the clinical relevance of the study, so from my point of view, not considering them could affect the clinical significance of its results. The authors mention that the points raised have not been considered in the REC approved protocol, so they do not consider it necessary to include them in this manuscript.

At this moment I do not have any other comment to share.

7. PLOS authors have the option to publish the peer review history of their article (what does this mean?). If published, this will include your full peer review and any attached files.

Reviewer #1: No

Reviewer #2: **Yes: **Ioannis Alagkiozidis

Reviewer #3: No

---

## [Editor Report · Acceptance letter]

7 Mar 2024

PONE-D-23-30937R1 

PLOS ONE

Dear Dr. Northgraves, 

I'm pleased to inform you that your manuscript has been deemed suitable for publication in PLOS ONE. Congratulations! Your manuscript is now being handed over to our production team.

Kind regards, 

on behalf of

Dr. Daniele Ugo Tari 

Academic Editor

PLOS ONE